# Light Pollution Disrupts Seasonal Differences in the Daily Activity and Metabolic Profiles of the Northern House Mosquito, *Culex pipiens*

**DOI:** 10.3390/insects14010064

**Published:** 2023-01-10

**Authors:** Matthew Wolkoff, Lydia Fyie, Megan Meuti

**Affiliations:** Department of Entomology, College of Food, Agricultural, and Environmental Sciences, The Ohio State University, Columbus, OH 43210, USA

**Keywords:** diapause, artificial light at night, FlyBox, Bradford assay, vanillin assay, anthrone assay, SCAMP

## Abstract

**Simple Summary:**

The Northern house mosquito transmits West Nile virus and survives the winter by entering a state of dormancy called diapause. Light pollution has been shown to interfere with diapause initiation in this mosquito. The effects of light pollution on daily activity rhythms and metabolic products have not been thoroughly investigated in diapausing and non-diapausing Northern house mosquitoes. We found that light pollution affected mosquito activity levels and several metabolic products differently depending on photoperiod, indicating that light pollution may disrupt nutrient accumulation and possibly interfere with diapause initiation in this species.

**Abstract:**

The Northern House mosquito, *Culex pipiens*, is an important disease vector, and females are capable of surviving the winter in a state of overwintering diapause. This species’ diapause response has been extensively studied, and recent evidence suggests that the circadian clock is involved in measuring seasonal changes in daylength to initiate the diapause response. However, differences in the circadian activity of diapausing and non-diapausing *Cx. pipiens* have not been thoroughly investigated. Additionally, recent findings indicate that artificial light at night (ALAN) can disrupt mosquito diapause, potentially prolonging the mosquito biting season. We compared the circadian locomotor activity of mosquitoes reared in diapause-averting, long-day conditions and diapause-inducing, short-day conditions with and without ALAN to elucidate the interplay between circadian activity, diapause, and light pollution. We also uncovered metabolic differences between mosquitoes reared under diapausing and non-diapausing photoperiods with and without ALAN by measuring the concentration of protein, fructose, glycogen, water-soluble carbohydrates, and lipids. We found that ALAN exposure altered several diapause-associated phenotypes including slightly, but not significantly, increasing activity levels in short day-reared mosquitoes; and preventing some short day-reared mosquitoes from accumulating lipids. ALAN also significantly reduced glycogen and water-soluble carbohydrate levels in long day-reared mosquitoes. Based on our findings, light pollution may decrease insect fitness by perturbing metabolism, and may also impact several phenotypes associated with insect diapause, potentially extending the mosquito biting season and preventing insects in urban environments from overwintering successfully.

## 1. Introduction

Organisms keep track of time throughout the day via an entrainable, endogenous mechanism known as the circadian clock [1]. Daily timekeeping in insects and other animals is primarily achieved through the light-dependent transcription of several key genes and subsequent negative feedback loops, which constitute a primary feedback loop [2,3]. The circadian clock circuit is best studied in *Drosophila*, and comprises approximately 150 neurons (reviewed by [4]). These are divided into the lateral neurons, which regulate various daily activity rhythms [5,6,7]; and the dorsal neurons, which are responsible for detecting and responding to daily light:dark cycles [8,9] and maintaining endogenous rhythms in the absence of light signals [10]. Clock neurons then communicate with other nervous system structures and various tissue-specific peripheral clocks reviewed in [11] to coordinate physiological activities and behaviors such as locomotor activity [12]; pheromone release [13]; emergence [14]; as well as feeding and metabolism [15], including blood feeding in mosquitoes [16].

Diapause is a physiologically dynamic state of arrested development that allows insects to survive prolonged seasons when the environment does not support growth and reproduction [17,18]. Females of the Northern house mosquito, *Culex pipiens pipiens* (referred to as *Cx. pipiens* hereafter), enter an adult reproductive diapause in response to the short days and low temperatures that they experience during their larval and pupal phases [19,20,21]. Diapause in *Cx. pipiens* is characterized by low levels of juvenile hormone which arrest egg follicle development [22], and a suppression of insulin signaling that leads to the upregulation of the forkhead transcription factor, FOXO [23,24]. Diapausing females also upregulate genes involved in fatty acid synthesis [25] allowing them to garner high levels of fat reserves. Behaviorally, diapausing females eschew imbibing vertebrate blood [26] and feed exclusively on nectar, often for prolonged periods throughout the night [27].

Although it is not entirely clear how *Cx. pipiens* distinguish long summer days from short winter days to initiate their adult, reproductive diapause, recent evidence suggests that the circadian clock is involved [28]. While the clock’s main role is timekeeping over a 24 h period, it has long been hypothesized that the circadian clock in plants and animals plays an integral role in detecting the seasonal changes in daylength that initiate diapause [29,30], reviewed by [31]. Diapause onset is triggered by short photoperiods in the autumn, which provide a highly reliable indicator of seasonal progression [20,32]. Diapause has independently evolved many times across all major insect taxa [33]; and has enabled insects to colonize temperate climates and exploit seasonally available resources [34].

Locomotor activity patterns can provide an indirect way to measure circadian rhythmicity [35], and studies in other taxa have leveraged this tool to assess the activity rhythms of diapausing and non-diapausing insects. Socha and Zemek [36] measured the locomotor activity of the linden bug, *Pyrrhocoris apterus* (Linnaeus) (Heteroptera: Pyrrhocoridae) and found that diapausing bugs moved significantly less than non-diapausing bugs. Diapausing females of the fruit fly *Drosophila triauraria* also exhibited reduced locomotor activity relative to non-diapausing flies [37]. Despite the widespread use of *Cx. pipiens* to study seasonal timekeeping, the locomotor activity patterns of diapausing and non-diapausing mosquitoes have not yet been fully characterized. During spring and summer, *Cx. pipiens* is generally crepuscular in the absence of light pollution, with peak activity occurring during dawn and dusk and sustained lower levels of activity throughout the night. As summer transitions to fall, *Cx. pipiens* alters its activity pattern, initiating flight earlier in the evening when light is more intense [38]. Additionally, sugar-feeding activity is higher in diapausing mosquitoes than non-diapausing mosquitoes [27], while the reverse is true of host-seeking behavior [39].

In addition to its robust and well-explored diapause response, *Cx. pipiens* also merits study due to its role in disease transmission. Females of this species are the primary vector of West Nile virus (WNV) in the northeastern United States [40]; and WNV has rapidly become the most prevalent mosquito-borne disease in the U.S. since its introduction in 1999 [41]. WNV has been isolated from overwintering mosquitoes [42,43,44], and WNV-infected females may reinitiate the cycle of disease transmission in the spring.

Light pollution, caused by anthropogenic Artificial Light at Night (ALAN), significantly interferes with both daily and seasonal activity patterns in insects and other animals [45]. The proliferation of anthropogenic light sources disrupts natural light cycles from both direct illumination and upward light scattering, known as skyglow [46]. The prevalence and intensity of ALAN globally has dramatically increased, and is now nearly ubiquitous as over 80% of the world experiences some level of light-polluted skies [47]. As light has been a major and reliable cue to synchronize daily and seasonal biological processes throughout the course of evolution, ALAN can significantly impede the function of those processes [48,49,50]. ALAN increases nighttime activity in primarily diurnal insects including endoparasitoid wasps [51] and the yellow fever mosquito, *Aedes aegypti*, such that ALAN causes these mosquitoes to blood-feed at night [52]. Although ALAN has profound effects on diurnal insects, nocturnal and crepuscular arthropods are particularly vulnerable to changes to their circadian behaviors that may be influenced by ALAN [53]. ALAN inhibits and generally decreases nighttime activity levels in nocturnal isopods, crickets, and amphipods [54,55,56]. Honnen et al. [57] found that in *Cx. pipiens f. molestus*, a sub-species that does not enter diapause, exposure to several hours of extra light during the dark phase of a normal light:dark cycle (also known as scotophase) results in depressed locomotor activity in this crepuscular mosquito. The effect of ALAN on phenology and other elements of seasonal biology in insects has been less studied. Studies have shown that ALAN can inhibit diapause initiation in flesh flies [58], leafminers [59], moths [59], and in mosquitoes including the Asian tiger mosquito, *Aedes albopictus* [60]. Our previous work also demonstrates that ALAN inhibits diapause entry in *Cx. pipiens* [61]. However, the effect of ALAN on the locomotor activity patterns in any diapausing insect has not been investigated.

In this study, we sought to characterize seasonal differences in the locomotor activity patterns of *Cx. pipiens*, and how light pollution might affect these systems. As previous studies demonstrate that diapausing *Cx. pipiens* spend more time sugar-feeding [27,62], and that ALAN disrupts seasonal differences in fat accumulation [61], we also investigated how differences in feeding and locomotor activity between diapausing, non-diapausing and ALAN-exposed mosquitoes affect their metabolic content. We found that non-diapausing *Cx. pipiens* were more active, and accumulated more glycogen and water-soluble carbohydrates than diapausing mosquitoes. In short day-reared mosquitoes, exposure to ALAN slightly increased daily activity levels and was associated with increased variability in lipid stores while ALAN suppressed glycogen and water-soluble carbohydrate accumulation in long day-reared mosquitoes. Our results demonstrate a photoperiod-dependent effect of ALAN that may perturb seasonal metabolic differences in *Cx. pipiens* and have important implications for their fitness, overwintering survival and ability to transmit disease.

## 2. Materials and Methods

### 2.1. Insect Rearing

Mosquitoes were obtained from the Buckeye strain, a laboratory colony of *Cx. pipiens pipiens* established in 2013 from egg rafts collected in Columbus, OH. To increase genetic diversity, adults originating from field-collected *Cx. pipiens* egg rafts were allowed to mate with Buckeye colony adults of the opposite sex in the summer of 2021. In the summer of 2022, several cages of Buckeye colony adults were fed chicken blood using an artificial membrane system (Hemotek, Blackburn, UK, SP4W1–3). The resulting first-instar larvae were reared to pupation in clear plastic containers (250 larvae per container) filled with ~750 mL of reverse-osmosis water, and fed ground fish flake (Tetramin, Tetra, Mulle, Germany). Within 24 h of pupation, approximately 150 pupae were moved to 8 oz plastic deli cups, and placed in clear plastic cages (30 cm × 20 cm × 15 cm).

Mosquitoes at all life stages were maintained in environmental chambers at 18 °C under diapause-inducing (8 h light, 16 h dark) or diapause-averting (16 h light, 8 h dark) photoperiods, with and without ALAN. Previous work has established that these photoperiod and temperature regimes consistently induce diapause or cause mosquitoes to avert diapause, respectively [20,21,62].

To minimize generational differences, larvae from a single batch of eggs were used to establish each long-day and short-day cohort. Long-day larvae were initially reared under long-day conditions (L:D 16:8) at elevated temperatures (24 °C) for 3 days post-hatching to accelerate their development. They were then moved to an environmental chamber with the same photoperiod (L:D 16:8) set to 18 °C. The long day-reared pupae thus emerged 5 days before the short-day reared pupae, ensuring adequate time for behavioral recording in the Fly Box system. Developing *Cx. pipiens* are particularly sensitive to diapause induction cues from the fourth larval instar through pupation [19,20,62]. At 24 °C, larvae of *Cx. pipiens* will, on average, require a minimum of 5 days to reach the fourth instar [63]. We therefore do not expect that exposing the first two larval instars to elevated temperatures to significantly impact the behavioral or metabolic phenotypes of the adult females utilized in subsequent experiments. All growth chambers were equipped with light fixtures which produced ~19,000 lux of light during the day. ALAN LED fixtures were constructed using a soft white LED lightbar partially covered with tape to reduce light intensity to 4 lux as measured by an Extech light meter (Extech LT300, Waltham, MA). This amount of artificial light is consistent with light pollution measured in suburban areas [64] including those of Columbus, OH [65], and which we have previously utilized to measure the effects of ALAN on *Cx. pipiens* [61]. Additional ALAN LED fixtures were constructed for a second smaller growth chamber and the Fly Box recorder using soft white LED puck lights (Lighting EVER, 1800016-DW-US), that were spray-painted black and sanded until 4 lux output was measured at a distance of 30 cm. ALAN lights were turned on throughout scotophase, and positioned within the environmental chambers such that all cages and larval rearing pans would have approximately equal exposure to ALAN (~4 lux). All larval pans, pupal cups, and adult cages were made of translucent plastic to allow maximum light penetration.

### 2.2. Fly Box Construction

The Fly Box activity monitor was constructed using a modified design of da Silva Araujo et al. [66]. An infrared LED floodlight (CMVision, CM-IR130-198 Illuminator) was used to illuminate the mosquito plate from below, and placed in a separate, ventilated compartment to minimize the amount of heat generated in the mosquito housing. A Logitech webcam was modified to remove the infrared filter and mounted to the lid of the activity monitor. A clear acrylic panel situated above the infrared floodlight served as the base for the 24-well mosquito plate.

### 2.3. Measuring Locomotor Activity

Approximately two days after adult eclosion, females were removed from the cage with an aspirator, chilled at −20 °C for 2 min, and kept in a plastic container on ice to maintain cold anesthetization. Cold-anesthetized females were individually placed in each well of a clear, 24-well polystyrene plate (12.5 cm × 8.5 cm × 2 cm, Lichen Cottage, EAN 8434529801091). To provide food and moisture during data collection, a 2 mm thick piece of cotton dental wick saturated with 10% sucrose solution was adhered to the sidewall of each well of the plate. A piece of clear plastic wrap placed over the plate, and six holes were poked above each well to provide ventilation. Females were held at room temperature for several minutes to ensure that they recovered from anesthetization. The entire plate was then placed in the Fly Box activity recorder. One day prior to the initiation of the continuous darkness (D:D) period, an additional 100 µL of 10% sucrose solution was added to each cotton wick using a micropipette.

All light fixtures were used in conjunction with programmable digital outlet timers (BN-Link, FD-60/U6). Starting from Zeitgeber 0 (the lights-on point of the first experimental day), mosquitoes were subjected to their normal light:dark cycle (16:8 for long-day mosquitoes, 8:16 for short-day mosquitoes) for 24 h to record their diel activity rhythms in response to exogenous photoperiodic cues. After this time, all lights were turned off, and their activity monitored for 48 h under continuous dark (D:D) conditions to record their endogenous circadian activity rhythms in the absence of exogenous photoperiodic input.

All data were collected using an Asus tablet. WebcamImageSave v1.11 was used to capture an image of the mosquito plate every 60 s. Following data collection, the plate was removed from the Fly Box, the mosquitoes were euthanized at −80 °C, and each female was stored in a microcentrifuge tube at −20 °C for metabolic analysis.

Image files were processed for analysis according to da Silva Araujo et al. [66]. All image files were loaded into PySolo-Video [67]. A configuration file was generated to calculate the distance each mosquito traveled, with tracking, during the experiment. A mask file was created by drawing bounding boxes around each well of the plate. Pvg-Acquire [67] was run using these configuration and mask files and the resulting text file was manually formatted for DAM analysis. DAMFileScan (TriKinetics Inc.) was used according to Vecsey [68] to sum activity counts for each mosquito into 1- and 30 min bins for analysis.

### 2.4. Metabolic Assays

Metabolic assays [69], modified by S. Wilson for use in *Cx. pipiens*, were used to measure protein, fructose, water-soluble carbohydrates, glycogen, and lipid levels in individual female mosquitoes whose behavioral activity had been monitored in the Fly Box. Following euthanasia, female mosquitoes were weighed. Next, 180 µL of buffer solution (100 mM KH_2_PO_4_, 1.0 mM dithiothreitol, 1.0 mM EDTA, adjusted to pH 7.4 using KOH) was added to each sample tube, and the mosquito was homogenized using an ultrasonic tissue homogenizer (BioLogics model 3000). As our largest mosquito weighed 5.44 mg, identical buffer volumes were used across all mosquito samples in accordance with the Foray et al. [69] protocol, which is suitable for use with insects weighing up to 15 mg. Protein levels from individual females were measured from this homogenate solution using a Bio-Rad Protein Assay Kit I (Bio-Rad, 5000001). Following protein quantification, 20 µL of 20% sodium sulfate, 10 µL of potassium dihydride phosphate, and 1500 µL of chloroform-methanol solution (1:2 *v/v*) were added to the homogenate. Homogenate solution was vortexed and centrifuged for 15 min at 180× *g*; and the resulting supernatant was used for fructose, carbohydrate, and lipid determination, while the pellet was used to measure glycogen. Fructose was measured using a cold anthrone test, while total water-soluble carbohydrates and glycogen were measured using separate hot anthrone tests on different fractions of the homogenized mosquitoes. Total lipids were measured using a vanillin assay. The metabolite extracts from each female mosquito were run in triplicate on a 96-well plate (MultiScreen, Sigma-Aldrich, MSPNNFX00). Concentrations of all metabolic products were measured using a FLUOstar Omega microplate reader (BMG Labtech).

### 2.5. Data Analysis

Analysis of circadian locomotor activity was performed using the Sleep and Circadian Analysis MATLAB Program (SCAMP; [68]). Actograms and mean activity counts were calculated for each experimental group. Chi-square periodograms and autocorrelation plots were used to determine activity periods and their associated statistical robustness, as indicated by the height and width of the main activity peak; and the rhythmicity statistic, respectively. The Chi-square periodogram estimates the period of time-series data by plotting Chi-square distributions against a 95% confidence interval, with the highest significant peak corresponding to the estimated period of the dataset. Significant rhythmicity statistics (>1) result when a set of time series data conforms closely (>95% CI) to a cosinor model, indicating robust circadian rhythmicity [37]. Autocorrelation analysis also provides an estimate of the periodicity in circadian data, by comparing correlation coefficients in a progressively time-shifted dataset. The period of the circadian rhythm is defined as the point in the time-shifted dataset at which the correlation coefficient achieves the highest peak, typically ~24 h for most insects [70].

The mean activity levels of each mosquito, calculated as a function of pixels traveled throughout the recording period, were tested for normality using an Anderson–Darling normality test and were found to be non-normal, (*p* < 0.05). Therefore, activity levels were normalized using a Box Cox transformation (*MASS* package, v.7.3-53, [71]). The normalized dataset was analyzed in R using linear mixed-effects models (*lme4* package, v1.1-31 [72]), with ALAN and photoperiod as fixed effects and cohort as a random effect. Each experimental replicate was considered as a distinct cohort for data analysis. Pairwise comparisons were made using general linear hypotheses with Tukey contrasts, and a Type III ANOVA was performed to assess the independent and combined effects of photoperiod and ALAN exposure. Lastly, the adjusted intraclass correlation coefficient (ICC) was calculated to estimate the proportion of variation explained by differences between cohorts (*sjstats* package, v0.18.2 [73]).

For the metabolite analyses, first the average absorbance and standard deviation of the three technical replicates for each mosquito were calculated. The abundance of each metabolite within an individual mosquito was calculated using standard curves with known levels of each metabolite. The relative abundance of all metabolites except lipids was then normalized to the fresh mass of the mosquito (ug metabolite/mg fresh mass); while lipids were normalized to mosquito lean mass (e.g., µg of mosquito fresh mass—µg of lipid [74]). Initial boxplots were used to identify outliers based on the interquartile range criterion, which were then removed on the basis that they likely resulted from errors during the assay protocol (e.g., metabolite levels fell outside of the range represented in the standard curves). The relative abundance of each metabolite was then tested for normality in R using an Anderson–Darling test. Non-normal datasets were transformed using a Box Cox transformation (*MASS* package, v.7.3-53, [71]), analyzed in R using linear mixed-effects models with ALAN and photoperiod as fixed effects and cohort as a random effect (*lme4* package, v1.1-31 [72]), followed by a Type III ANOVA and intraclass correlation coefficient (ICC) calculation to estimate the proportion of variation explained by cohort differences (*sjstats* package, v0.18.2 [73]).

Boxplots were created using unnormalized activity and metabolite data, in order to show the distribution of sample data for each experimental measure. Effects plots were generated in R (*effects* package, v.4.2-2 [75,76] using normalized data to more clearly illustrate how activity and metabolite levels were affected by photoperiod in the presence and absence of ALAN.

## 3. Results

### 3.1. Survival

Of the 24 mosquitoes placed in the well, 22.4 ± 1.08 (mean ± SD) per plate survived to the end the activity recording period. Mosquitoes that perished during the experiment, as indicated by a permanent lack of recorded activity, were excluded from all subsequent behavioral and metabolic analyses.

### 3.2. Activity Analysis

Activity patterns differed significantly between long-day reared, non-diapausing (LD) and short-day reared, diapausing (SD) mosquitoes. LD mosquitoes were generally crepuscular, with large activity peaks occurring at the onset of both scotophase and photophase, with moderate levels of activity throughout scotophase and low levels of activity during photophase (Figure 1A). These patterns were preserved in constant darkness (D:D), indicating that these activity patterns were maintained by the mosquitoes’ endogenous circadian clock. SD mosquitoes, in contrast, exhibited sustained low levels of activity throughout the night (Figure 1B). These behavioral patterns were also preserved in D:D conditions. Furthermore, SD mosquitoes exhibited significantly lower mean activity levels than LD mosquitoes (Figure 2A, Appendix A, Tukey’s HSD, z = −3.898, *p* < 0.001).

Similar to LD mosquitoes reared without ALAN, the activity patterns of LD mosquitoes reared with artificial light at night (LD + ALAN; Figure 1C) had peak activity occurring at the onset of the scotophase and photophase in two of the three cohorts. All LD + ALAN mosquitoes in two of three cohorts exhibited activity peaks at the onset and conclusion of scotophase, while the mosquitoes in the third cohort exhibited an aberrant pattern of high activity during subjective photophase and reduced activity during scotophase (Figure 1C).

ALAN did not significantly influence locomotor activity levels in LD + ALAN mosquitoes compared to LD mosquitoes (Figure 2A, Appendix A, Tukey’s HSD, z = 1.220, *p* = 0.614). SD + ALAN mosquitoes did not exhibit consistent activity peaks at the onset of scotophase or photophase, instead showing prolonged periods of elevated activity throughout scotophase in L:D conditions and subjective scotophase in D:D conditions (Figure 1D). SD + ALAN mosquitoes exhibited higher levels of activity than SD mosquitoes, which approached the level of significance (Figure 2A, Appendix A, Tukey’s HSD, z = −2.393, *p* = 0.078).

ANOVA performed on the linear model for activity level indicated that neither photoperiod (Type III Wald Test, *X*^2^ = 0.079, *p* = 0.779) nor light pollution (Type III Wald Test, *X*^2^ = 1.487, Df = 1, *p* = 0.223) was found to independently influence mosquito activity level, while a significant interaction between photoperiod and light pollution was detected (Type III Wald Test, *X*^2^ = 6.530, Df = 1, *p* = 0.011; Figure 2A, Appendix A). Cohort as a random effect was found to significantly influence the model (REML Likelihood Ratio test, LRT = 22.706, Df = 1, *p* < 0.001), with 17.9% of variation in activity level explained by variation between cohorts (adjusted ICC = 0.179).

### 3.3. Circadian Periodicity

All experimental groups exhibited circadian activity periods in constant darkness that slightly exceeded 24 h based on both periodogram and autocorrelation analyses. LD mosquitoes exhibited a circadian period between 26.5 and 27.8 h for autocorrelation and periodogram analysis, respectively; SD mosquitoes between 25.7 and 27 h; LD + ALAN mosquitoes between 26 and 26.6 h; and SD + ALAN mosquitoes between 25.5 and 25.8 h. Autocorrelation rhythmicity statistics were significant (RS > 1) for all cohorts/trials except one (SD18-Cohort B, RS = 0.6); while Chi-square periodogram analysis identified significant peaks (>95% confidence limit) in all cohorts/trials except one (SD18 + ALAN-Cohort B). No significant differences in circadian period were identified between the experimental groups, indicating that neither photoperiod nor ALAN exposure significantly affected the periodicity of mosquito circadian rhythm.

### 3.4. Metabolic Analysis

Photoperiod and ALAN were found to significantly influence the levels of several metabolic products (Figure 2B–F), with photoperiod-dependent effects on metabolite level (Appendix A). Protein levels were unaffected by photoperiod (Type III Wald Test, *X*^2^ = 1.187, Df = 1, *p* = 0.276) or light pollution (Type III Wald Test, *X*^2^ = 0.201, Df = 1, *p* = 0.654); combined, there was not a signfiicant interaction between these two variables (Type III Wald Test, *X*^2^ = 0.961, Df = 1, *p* = 0.327). No significant differences were identified between any of the experimental treatments, although one cohort of SD + ALAN mosquitoes exhibited elevated protein levels (data not shown). Relatedly, cohort was found to significantly influence the linear model (REML Likelihood ratio test, LRT = 12.175, Df = 1, *p* < 0.001), with 30.5% of the variation in protein levels explained by differences between cohorts (adjusted ICC = 0.305).

Fructose levels (Figure 2C) were not significantly effected by photoperiod (Type III Wald Test, *X*^2^ = 0.554, *p* = 0.457) or ALAN (Type III Wald Test, *X*^2^ = 0.008, *p* = 0.927), nor was any significant interaction measured between the two conditions (Type III Wald Test, *X*^2^ = 0.092, *p* = 0.762). Cohort was found to signficantly influence the model (REML Likelihood ratio test, LRT = 8.222, Df = 1, *p* = 0.004), with 34.1% of the variation in fructose levels explained by differences between cohorts (adjusted ICC = 0.341).

Photoperiod did not significantly influence levels of water-soluble carbohydrates (Figure 2D, Type III Wald Test, *X*^2^ = 0.293, *p* = 0.588), nor was a joint effect identified between photoperiod and ALAN (Type III Wald Test, *X*^2^ = 1.545, Df = 1, *p* = 0.214). However, ALAN alone was found to significantly affect total water-soluble carbohydrates (Type III Wald Test, *X*^2^ = 12.416, Df = 1, *p* < 0.001). ALAN significantly reduced water-soluble carbohydrates in LD mosquitoes relative to LD mosquitoes reared without ALAN (Tukey’s HSD, *z* = 3.524, *p* = 0.003). Cohort was also found to significantly affect the water-soluble carbohydrates model (REML Likelihood ratio test, LRT = 20.084, Df = 1, *p* < 0.001), with 43.4% of the variation in carbohydrate levels explained by differences between cohorts (adjusted ICC = 0.434).

Both photoperiod (Type III Wald Test, *X*^2^ = 4.835, Df = 1, *p* = 0.027) and ALAN (Type III Wald Test, *X*^2^ = 9.637, Df = 1, *p* = 0.002) exhibited significant effects on glycogen level; and a highly significant joint effect was also measured (Type III Wald Test, *X*^2^ = 12.222, Df = 1, *p* < 0.001). Cohort was also found to significantly influence the model (REML Likelihood ratio test, LRT = 12.398, Df = 1, *p* < 0.001), with 43.9% of the variation in glycogen level attributable to differences between cohorts (adjusted ICC = 0.439). In the absence of ALAN, SD mosquitoes exhibited lower glycogen levels (Figure 2E, Appendix A) than LD mosquitoes (Tukey’s HSD, *z* = −2.732, *p* = 0.031). ALAN substantially suppressed glycogen accumulation in LD mosquitoes (Appendix A, Tukey’s HSD, *z* = 3.104, *p* = 0.010), while ALAN increased glycogen accumulation in SD mosquitoes (Appendix A), although this effect was not significant (Tukey’s HSD, *z* = −1.990, *p* = 0.188).

Lipid levels were not significantly influenced by photoperiod (Type III Wald Test, *X*^2^ = 1.694, Df = 1, *p* = 0.193) or ALAN (Type III Wald Test, *X*^2^ = 0.408, Df = 1, *p* = 0.523); and there was not a significant interaction between these variables (Type III Wald Test, *X*^2^ = 0.110, Df = 1, *p* = 0.740). Additionally, cohort was found to significantly influence the lipid model (REML Likelihood ratio test, LRT = 16.191, Df = 1, *p* < 0.001), with 47.4% of the variation in lipid levels explained by differences between cohorts (adjusted ICC = 0.474). Although lipid levels were higher in SD mosquitoes (170.6 ± 19.1 ug lipid/mg mosquito lean mass) than LD mosquitoes (92.3 ± 8.14 ug lipid/mg mosquito lean mass), there were not any statistically significant differences in the lipid levels between these groups (Tukey’s HSD, *p* > 0.05).

## 4. Discussion

This study represents the first time that both circadian locomotor activity and metabolite levels have been assessed within individual diapausing and non-diapausing insects. We have additionally uncovered how ALAN alters daily behavioral activity and metabolic composition in different seasonal contexts. Previous studies have demonstrated that ALAN inhibits diapause initiation in insects [59,60]. Additionally, we have previously shown that ALAN allows short day-reared females of *Cx. pipiens* to accumulate body fat, but remain reproductively active [61]. The present study has further expanded on this body of work by showing that ALAN alters the levels of several other metabolites in a photoperiod-dependent fashion.

Both photoperiod and ALAN influenced circadian locomotor activity. As we hypothesized, long-day mosquitoes exhibited higher mean activity levels than short-day mosquitoes, possibly because diapausing mosquitoes reduce activity to conserve metabolic energy while overwintering [77]. There was, however, a significant interaction between photoperiod and light pollution which was likely caused by the non-significant decrease in the locomotor activity of LD + ALAN exposed mosquitoes relative to LD mosquitoes and the slight but non-significant increase the locomotor activity of SD + ALAN exposed mosquitoes relative to SD females. This suggests that the influence of light pollution on mosquito behavior is dependent on photoperiod.

Studies of other insects [54,55,56], including the closely-related subspecies *Cx. pipiens molestus* that is incapable of entering diapause [57], found that ALAN suppressed locomotor activity, which is inconsistent with our findings in LD, non-diapausing *Cx. pipiens*. One possible explanation is that our ALAN fixtures emitted at 4 lux, which may have been sufficiently dim to maintain normal flight activity in LD mosquitoes as Veronesi et al. [38] found that, in the spring and summer, *Cx. pipiens* initiated flight below 5 lux during early scotophase. *Cx. pipiens* may therefore be expected to maintain normal host-seeking and biting activity in suburban and urban areas despite moderate levels of light pollution.

Relatedly, the moderate increase in circadian locomotor activity observed in SD + ALAN mosquitoes suggests that *Cx. pipiens* may continue to engage in feeding behaviors such as host-seeking despite the presence of light pollution, which is consistent with our earlier findings [61]. Lastly, the disparate responses to ALAN observed in SD and LD mosquitoes may reflect seasonal differences in light sensitivity. While mosquitoes initiate evening flight below 5 lux in the spring and summer, in the fall *Cx. pipiens* initiate flight earlier in the evenings when light is more intense. Thus, the higher activity levels exhibited by SD + ALAN mosquitoes may reflect a shift in when diapausing mosquitoes initiate flight activity.

ALAN reversed seasonal changes in the abundance of several metabolites in a photoperiod-dependent fashion. Contrary to our expectations, protein levels were not significantly different between any of the experimental treatments (Figure 2B and Appendix A). Many insect species reduce their flight musculature during diapause to conserve energy [78,79], including *Cx. pipiens* [77]. Diapausing mosquitoes are also generally less active than non-diapausing mosquitoes [26]. Our findings of comparable protein levels between long-day and short-day mosquitoes conflict with these studies, as diapausing mosquitoes were expected to exhibit lower protein abundance as a consequence of reduced musculature. One possible explanation is that the upregulation of other proteins compensated for reduced muscle mass in short-day reared mosquitoes. Insect diapause is associated with the upregulation of storage hexamerins [80,81,82] and a diverse suite of enzymes [83], which may have led to increased protein abundances despite reduced musculature. Additionally, adult mosquitoes were maintained in a 24-well plate for 4 days, which may have led to general reductions in muscle mass due to disuse after being placed in a small enclosure. However, the relationship between disuse and atrophy in insect musculature is unclear. In *Drosophila*, neither winglessness nor prolonged paralysis was associated with flight muscle degeneration [84], although atrophy was identified in cockroach thoracic muscle, but only after 9 weeks of disuse [85].

While short-day mosquitoes did not demonstrate reduced protein abundance, higher mean protein levels were measured in one cohort of SD + ALAN mosquitoes relative to all other treatments (data not shown). This observation may warrant further investigation, as it could reflect a scenario in which ALAN exposure caused mosquitoes to maintain muscle mass to support flight activity, while still upregulating some diapause-associated proteins. High levels of activity are associated with increased muscle retention in *Drosophila* [86], and the slight increase in locomotor activity observed in SD + ALAN mosquitoes may therefore have caused some mosquitoes to maintain muscle mass. Such a phenotype would be consistent with our previous work, which found that short-day reared, ALAN exposed mosquitoes were able to garner the high levels of lipids associated with diapause, while preserving host-seeking and reproductive capabilities [61].

Bowen [27] observed more frequent sugar feeding in diapausing *Cx. pipiens* relative to non-diapausing females. Therefore, we hypothesized that short-day reared females would have higher fructose levels than LD and SD + ALAN mosquitoes. Contrary to these expectations, there were no significant differences in fructose levels (Figure 2C). However, our findings corroborated those of Zhou and Miesfeld [87], who did not find significant differences in glucose content between diapausing and non-diapausing mosquitoes. These authors postulated that diapausing mosquitoes may feed more frequently, but take smaller meals than non-diapausing mosquitoes. Another possibility is that diapausing females may more efficiently convert sugar meals into lipid and glycogen stores.

Contrary to our hypotheses, total water-soluble carbohydrate levels were significantly higher in LD mosquitoes than in SD mosquitoes (Figure 2D). Carbohydrate levels are generally higher in diapausing insects to provide energy reserves throughout winter [88]. Similarly, glycogen levels also were significantly higher in LD mosquitoes than in SD mosquitoes (Figure 2E). The observed discrepancies in carbohydrate and glycogen levels may be because only 7-day-old mosquitoes were used for metabolite analysis. Using radioactive carbon tracing, Zhou and Miesfeld [87] found that significant glycogen accumulation does not occur in diapausing *Cx. pipiens* mosquitoes until 9 days after adult emergence; prior to this time, non-diapausing mosquitoes exhibit slightly more glycogen. Our mosquitoes were therefore unlikely to have accumulated large quantities of glycogen and other carbohydrates prior to the metabolite analysis. Zhou and Miesfeld [87] also found that, unlike glycogen, SD mosquitoes accumulate significantly higher lipid stores within 6 days of eclosion, and during this period, diapause-destined mosquitoes converted 46% more glucose into lipid than non-diapausing mosquitoes. Therefore, the SD mosquitoes in our experiments may have exhibited reduced glycogen and carbohydrate content compared to LD mosquitoes because they prioritized lipid synthesis, causing these mosquitoes to convert a greater percentage of ingested sugars into fat during the study period. This may also explain why SD + ALAN mosquitoes had higher, though statistically insignificant, glycogen levels compared to SD mosquitoes. If ALAN suppressed diapause induction under SD conditions as has been observed in other insects [58,59,60], then SD + ALAN mosquitoes may have prioritized glycogen production over lipid synthesis, similar to LD-reared, non-diapausing mosquitoes.

ALAN suppressed the accumulation of both water-soluble carbohydrates and glycogen in LD mosquitoes (Figure 2D,E, Appendix A). In other insects, ALAN has been found to influence metabolism and feeding [51,89,90,91]. Low carbohydrate and glycogen levels in ALAN-exposed mosquitoes may therefore suggest that light pollution disrupts the ability of LD-reared mosquitoes to acquire carbohydrates.

Fat hypertrophy is a characteristic physiological change that accompanies diapause in *Cx. pipiens* [23,25,92]. Our previous study [61] found that females of *Cx. pipiens* reared under SD + ALAN conditions accumulated less lipid than diapausing females. We identified a similar, though non-significant pattern, in SD + ALAN mosquitoes (Figure 2F, Appendix A). Our results also indicated that exposure to ALAN increased the variation in the amount of lipids acquired by SD-reared females, such that SD + ALAN females had mean lipid levels comparable to LD mosquitoes. This may imply that some of these females have not fully committed to the diapause program, and would thus be unlikely to survive prolonged periods of food scarcity during winter.

In summation, these findings imply that light pollution causes LD-reared, nondipausing females to alter their metabolism or feeding behavior. While the mechanisms underpinning these differences are unclear, the insulin signaling pathway represents a potential candidate for future study linking ALAN to metabolic disruption. Insulin signaling is suppressed in diapausing females of *Cx. pipiens*, which relieves the inhibition of the forkhead transcription factor (FOXO) and promotes lipid synthesis [23,25]. The circadian transcription factor *Par-domain protein 1* (*Pdp1*) also appears critical to lipid synthesis in diapausing mosquitoes [93], and may serve as a link between the circadian and insulin signaling systems. ALAN has been shown to disrupt circadian signals in other insects [48]. Therefore, perturbations of the circadian clock caused by ALAN in *Cx. pipiens* may disrupt the insulin signaling pathway via *Pdp1* to suppress carbohydrate and glycogen accumulation.

One potential confounding factor in the present study is the use of increased rearing temperatures during the first 4 days of larval development in the LD experimental groups. While both LD and SD mosquito were provisioned with the same amount of food at all life stages, reared at the same densities, and reared under identical temperatures from the third instar onward, the possibility remains that increased temperatures during the first two instars may have permitted LD and LD + ALAN larvae to accumulate different nutrient levels. Some evidence suggests that this initial rearing difference may ultimately be insignificant to adult metabolic content. Timmerman and Briegel [94] found that over 84% of protein and lipid accumulation in *Cx. pipiens* larvae occurred during the fourth instar. Current evidence also suggests that diapause-associated fat hypertrophy and carbohydrate accumulation are largely predicated on sugar feeding by adult mosquitoes [27,94,95], rather than larval feeding. Conversely, rearing temperatures can impact adult body size in *Cx. pipiens* [63]. However, as we normalized the metabolite levels in each mosquito to body mass, it is unlikely that body size would have influenced our results.

Several exciting questions about how ALAN affects both daily patterns of insect activity and seasonal responses remain to be explored. Future studies should also measure the activity patterns and metabolic contents of older mosquitoes. Fat hypertrophy and glycogen accumulation are elevated within seven days post-eclosion [92], but may not peak until 10 or more days post-eclosion [87]. The timepoints used in our study may therefore have preceded peak lipid and glycogen abundance in SD mosquitoes. Our previous study showed that exposure to ALAN increased the blood feeding propensity and reproductive capacity of SD-reared mosquitoes [61], but it is currently unknown whether light pollution would similarly affect the blood-feeding and fecundity of LD-reared females. Lastly, future studies should employ different methods to identify the molecular underpinning of the observed phenotypes. For example, RNA-interference or quantitative PCR could be used to identify key genetic elements that are involved in regulating circadian activity and metabolism in diapausing and nondiapausing mosquitoes, and whether light pollution affects the abundance of their transcripts and/or proteins.

The photoperiod-dependent effects of light pollution may prove important when evaluating the impact of ALAN on insect fitness and disease transmission. Short day-reared mosquitoes exposed to ALAN are more active and have variable amounts of lipid, suggesting that they may retain some characteristics of nondiapausing mosquitoes such as blood-feeding. Conversely, light pollution reduced carbohydrate and glycogen stores in LD-reared mosquitoes and may lead to reduced fitness of mosquitoes in urban areas. Female mosquitoes that cannot acquire sufficient sugars are less fecund and do not live as long as well-nourished females [96], and pre-diapausing mosquitoes must gorge on sugar sources in order to accumulate the lipid stores necessary to survive the winter [92]. More broadly, our findings demonstrate that ALAN can significantly inhibit the synthesis or accumulation of key metabolic stores, and suggest a mechanism by which light pollution could be contributing to global declines in insect populations [97].

## 5. Conclusions

This work characterized, for the first time, the circadian locomotor activity of *Cx. pipiens* under both diapause-averting and diapause-inducing conditions. Additionally, we have uncovered how both photoperiod and light pollution affect circadian activity and the abundance of several key metabolic products. In long day-reared mosquitoes, ALAN led to a slight, but non-significant decrease in activity levels and significantly suppressed water-soluble carbohydrate and glycogen levels. In short day-reared mosquitoes, light pollution led to slight, statistically insignificant increases in locomotor activity and protein levels in female mosquitoes, while increasing lipid variability. These results, combined with our earlier study [61], suggest that light pollution causes female *Cx. pipiens* to avert diapause. This could extend the mosquito biting season and increase mosquito-borne disease transmission during autumn within urban areas. Conversely, our results suggest that during the summer, light pollution disrupts the accumulation of essential nutrient stores and may decrease the fitness of mosquitoes and other insects.

## Figures and Tables

**Figure 1 insects-14-00064-f001:**
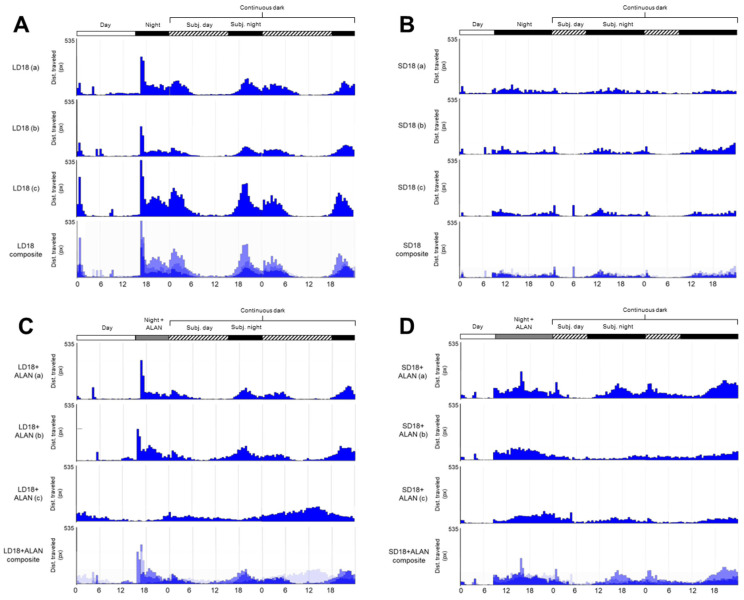
Photoperiod and light pollution influence mosquito circadian activity. Locomotor activity of *Cx. pipiens* mosquitoes reared under (**A**) long-day, diapause averting conditions (LD); (**B**) short-day, diapause-inducing conditions (SD); (**C**) long-day conditions with artificial light at night (LD + ALAN); and (**D**) short-day conditions with artificial light at night (SD + ALAN). White bars, labeled “Day”, indicate when lights were on; black bars, labeled “Night” and “Subj. night” (Subjective night) during the continuous dark period, indicate when lights were off; gray bars, labeled “Night + ALAN”, indicate when light was off, but ALAN was present; and hashed bars, labeled “Subj. day” indicate the subjective day in continuous dark conditions. Each graph represents the average activity of 20–24 female mosquitoes from one experimental trial or cohort, with a total of three cohorts for each treatment. The fourth actogram in each set is a composite of the preceding three demonstrating the consistency of activity patterns across experimental trials.

**Figure 2 insects-14-00064-f002:**
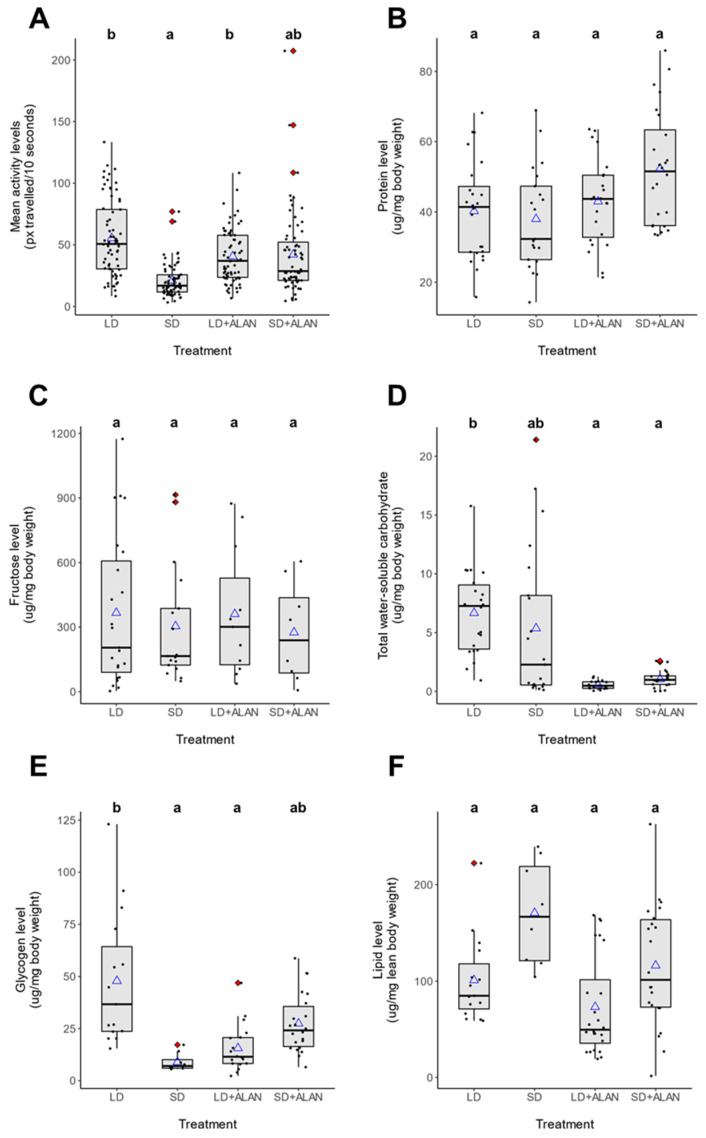
Photoperiod and light pollution influence seasonal differences in mosquito circadian activity and the levels of metabolites. (**A**) Activity level in LD-reared mosquitoes was significantly higher than SD-reared mosquitoes. Exposure to ALAN resulted in a slight activity increase in SD mosquitoes which approached significance (Tukey’s HSD, z = −2.393, *p* = 0.078). (**B**) Protein levels did not differ significantly among any experimental groups. (**C**) Fructose levels did not vary across different treatments. (**D**) SD mosquitoes exhibited lower levels of water-soluble carbohydrate than LD mosquitoes. ALAN reduced water-soluble carbohydrate levels in both LD and SD mosquitoes. (**E**) Glycogen levels were significantly higher in LD mosquitoes than SD mosquitoes. Exposure to ALAN reduced glycogen levels in LD mosquitoes. In contrast, ALAN increased glycogen levels in SD mosquitoes, but these results were not statistically significant. (**F**) SD mosquitoes exhibited higher lipid levels than LD mosquitoes, but these results were not statistically significant. ALAN did not significantly affect lipid levels in LD or SD mosquitoes. Black circles represent individual mosquitoes in each trial; blue triangles represent means; and red diamonds represent statistical outliers. Letter codes were used to indicate groups which were significantly different (Tukey’s HSD, *p* < 0.05).

## Data Availability

All data have been made available on the FigShare data repository. Mean locomotor activity data are available from DOI: 10.6084/m9.figshare.21533838. Monitor files generated by PySolo-video from mosquito recordings are available from DOI 10.6084/m9.figshare.21533961. Metabolic assay data, including raw absorbance readings, standard curves, and analysis-ready CSV files are available from DOI 10.6084/m9.figshare.21533976. FlyBox construction details and schematic are available from DOI 10.6084/m9.figshare.21534321.

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
