# Peer review of "Light Pollution Disrupts Seasonal Differences in the Daily Activity and Metabolic Profiles of the Northern House Mosquito, Culex pipiens"

_insects, 2023, doi:10.3390/insects14010064_

Round 1
Reviewer 1 Report
This is a strong and clear paper that adds to the growing body of work on artificial light at night and how it can disrupt physiological processes in insects. The focus here is on effects on activity levels and energetic reserve accumulation by Culex pipiens.
Besides a few typos, and an empty bracket where presumably statistical test information was intended (l367), I only have one concern for the authors to address, and that is about how they addressed their replicates in the statistical analysis. It is not entirely clear if the results are presented for each replicate / cohort individually (some figures do, but the results section does not seem to), whether cohort was included as a random factor in the analysis, or whether any statistical differences among cohorts were found to be absent.
Where this potentially matters is in Fig 2 for protein levels for the SD+ALAN treatment, where we see a wide spread of values. Do those represent different cohorts, or is there some other aspect about mosquitoes in this treatment that led to this bivariate distribution? I also missed discussion about this pattern.
Author Response
Thank you very much for taking the time to review our manuscript, and for your helpful input. Please see the attachment for a point-by-point response to your comments.
Reviewer 2 Report
Please find my comments and suggestions in the attached Word document.

Author Response

(The authors gave the same response as above.)

Reviewer 3 Report
The manuscript describes experiments studying the effects on artificial light on Culex pipiens mosquitoes in both a long-day setting as well as in a short-day setting. While interesting the study stops short of investigating several interesting mechanisms that are discussed. The paper would benefit from more functional assays into the mechanisms that control the ALAN effects.
The species name of the mosquito studied should be in the title, the studied effect might not be applicable to more species than Culex pipiens that diapauses.
In the abstract, line 23, both long-day conditions and short-day conditions are described as diapause averting.
The introduction summarizes the results in an unnecessary way.
introduction line 123 page 3, “and” twice.
In the methods, line 149 inches are used. Instructions for authors state: SI Units (International System of Units) should be used. Imperial, US customary and other units should be converted to SI units whenever possible.
Please discuss differences in light regimens when results from different papers are compared. ALAN studies use a wide variety of settings, how did you choose the settings you use?
You use mosquitoes that are a cross between a lab strain (buckeye) and wild caught mosquitoes, Why did you choose this setup?
It might be of interest to make sure that you do not introduce Culex pipiens f molestus mosquitoes with completely different patterns where they do not diapause.
How do you blood feed the strain? This is missing from the methods.
Discussion mentions several physiological changes that could have been measured.
“Our findings of comparable protein levels between long-day and short-day mosquitoes conflict with these studies, as diapausing mosquitoes were expected to exhibit lower protein abundance as a consequence of reduced musculature. One possible explanation is that the upregulation of other proteins compensated for reduced muscle mass in short-day reared mosquitoes. Insect diapause is associated with the upregulation of storage hexamerins [73–75] and a diverse suite of enzymes [76], which may have led to increased protein abundances despite reduced musculature.”
I would like to see experiments addressing this either by looking at transcription or through specific assays targeting these proteins.
“perturbations of the circadian clock caused by ALAN in Cx. pipiens may disrupt the insulin signaling pathway via Pdp1, possibly reducing carbohydrate, glycogen, and lipid accumulation.”
Also this speculation could be addressed through rtPCR or other specific assays.
Author Response

(The authors gave the same response as above.)

Reviewer 4 Report
This manuscript describes the activity and metabolic levels of long-day and short-day Cx. pipiens exposed to artificial light at night (ALAN). The authors provide strong evidence for the ALAN changing the activity levels of both long day and short day conditions.
I think the manuscript is well written and care was taken in its preparation. I would like to see a few small areas addressed (1) adding a line or two of text that explains the decision not to adjust the volume of buffer used based on mosquito mass when preparing the homogenates for metabolize analysis. (2) adding a section to rationalize the collection of the data in dark-dark conditions. The data show one 24 h cycle of light-dark before switching to dark-dark for 48 h. I feel it would be more compelling to see a greater number of standard cycles of light-dark. (3) a sentence to address the use of 24Ëš C to accelerate the growth of the long-day mosquitoes. Is there a study that has demonstrated this to have no dominant effect?
Minor points to improve this work.
Line 11-14: This sentence reads, "We found… in this and other insect species." No other insects were included in this study, so the other insects should be removed.
Line 23: Diapause inverting, short day conditions or diapause inducing?
Line 123: levels is misspelled.
Line 276: Cx. pipiens needs to be italicized
Line 285: Figure is missing the A, B, C, D, E, F designations
Line 352: letters in figure indicate that photoperiod is significantly different. This needs to be reflected in the text and figure need to be brought into alignment.
Author Response

(The authors gave the same response as above.)

Round 2
Reviewer 2 Report
Dear Authors,
Thank you very much for revising your manuscript! I do find your statistical results and methods much more appropriate. However, I still have a number of comments and suggestions to your manuscript:
I would in general argue that you should only focus on results that are statistically significant. There might be differences between groups in your sample of mosquitoes, but if those differences are not statistically significant, then they might be caused by random chance, and not a real difference in the mosquito populations for which your mosquitoes are a representative sample of.
I very much appreciate you making and adding the effects plots. I would argue that they should replace Figure 2, perhaps with adding the data points on top of the effects plots using ggplot.
Here are my specific comments and suggestions:
1. In the Abstract, line 26, you mention three groups of mosquitoes (diapausing, non-diapausing and ALAN-exposed). However, reading the paper it is clear that there were 4 groups of mosquitoes, both diapausing and non-diapausing with and without ALAN exposure. Can you make sure this is clear in the Abstract?
2. Now I understand that the amount of light during rearing determines the diapausing status of mosquitoes, without having to check the egg follicles. Would it be then simpler to just say either lon-day reared and short-day reared, or diapausing vs non-diapausing throughout the manuscript, instead of using both? What is the benefit of identifying them via both diapausing vs non-diapausing, and also short-day reared vs long-day reared, instead of just using diapausing vs non-diapausing, without referring to the long-day vs short-day rearing continuously?
3. The sentence in the Abstract starting "We found... " on line 27 is based on non-significant differences between short-day reared mosquitoes with and without exposure to ALAN. You do say "slightly", but I'd like you to state "but not significantly" to make clear that these results are not statistically significant.
4. On line 124, it isn't clear if the differences between diapausing and non-diapausing mosquitoes are present in mosquitoes in the presence of or in the absence of ALAN, or both? I would assume this is in the absence of ALAN, but please state that explicitely.
5. On line 139, change "diveristy" to "diversity".
6. On line 162, change "significantly impacted" to "to significantly impact".
7. On line 163, delete "used".
8. The paragraph on lines 177-186 feels like it belongs later in the Methods, such as in the section 2.4, instead of where it is now at the end of the section on Insect rearing.
9. On line 273, you state the you used a "generalized linear mixed-effects model". Was it really generalized, not just general? Did you use a family of distribution for the response variable other than normal? Same thing on line 291 and line 332.
10. On line 324, please change "LD+Ctrl" to just "LD".
11. On line 325, you state that "However, ALAN altered the activity levels of SD-reared mosquitoes (Figure 1D)." Is this difference statistically significant? Not based on Figure 3A! In the previous line, you talk about the same for LD mosquitoes, but refer to Figure 3A. I would suggest to keep this sentence to Figure 1, and talk about that with LD mosquitoes only in the previous sentence.
12. On line 326 and 367, change "SD18+ALAN" to "SD+ALAN".
13. In lines 332-335, you talk about how neither photoperiod nor light pollution significantly impacted activity levels, but there was a significant interaction. In terms of interpreting this, I would add that this indicates that the effect of ALAN is significanlty different in LD vs SD mosquitoes, or the effects of LD vs SD is significantly different in the absence or presence of ALAN. We can see that ALAN significantly decreases activity in LD mosquitoes, but not in SD mosquitoes. You could also say that while in the absence of ALAN, LD mosquitoes are significantly more active compared to SD mosquitoes, that difference is no longer significant in the presence of ALAN, suggesting that ALAN removes the difference between LD and SD mosquitoes by significantly reducing activity in LD mosquitoes, and slightly but not significantly increasing activity in SD mosquitoes. Do I interpret your results correctly?
14. On line 336-338, you state that cohort as a random effect was significant. This is useful as it suggests that mosquitoes in different cohorts behaved differently. However, it would be useful to know what percentage of the variation in the response variable is explained by the variation between cohorts as opposed to by variation overall among mosquitoes. This is called the intraset correlation coefficient, and can be calculated based on the output of your mixed effects model. The icc() command can do that in the sjstats package. You can read more about it here: https://cran.r-hub.io/web/packages/sjstats/vignettes/mixedmodels-statistics.html. I'd like to ask for the same thing for the other mixed effects models for the metabolic products.
15. On line 378-379, you state that one cohort of SD+ALAN mosquitoes exhibited elevated protein levels and refer to Figure 2B. However, I can't see that on Figure 2B, since all the three cohorts are denoted by black circles. If you really would like to show that, you could color the circles by cohort. Alternatively, you could just say that "data not shown" instead of Figure 2B. The same issue is raised on line 494 in the Discussion.
16. On line 386-388, you state that "the effects plot indicated a very slight, statistically insignificant decrease in fructose levels in...SD conditions...". Looking at Figure 3C, I don't see this, and I'm not convinced that what you're saying is real. I would suggest removing this conjecture.
17. On line 397, you state that "While carbohydrate levels were reduced in SD mosquitoes relative to LD mosquitoes...". What is this statement based on? Figure 3D indicates that in the absence of ALAN, there is no significant difference between LD and SD mosquitoes. Based on this, I'm not convinced of your following statement on how ALAN negated this pattern, since the pattern was not significant to begin with. Please remove this conjecture.
18. On line 406, you state that "SD mosquitoes exhibited lower glycogen levels ... than LD mosquitoes". I would suggest to make this more specific by saying "In the absence of ALAN, SD mosquitoes exhibited significantly lower glycogen levels ... than LD mosquitoes."
19. In the caption of Figure 2, line 429, you state that "D) SD mosquitoes exhibited lower level of water-soluable carbohydrate than LD mosquitoes." This is certainly true for the mosquitoes you tested, but this is not statistically significant difference based on Figure 3D. Please acknowledge that here. The same is true for the effect of ALAN on SD mosquitoes on water-soluble carbohydrate levels, and the effect of ALAN on glycogen levels in SD mosquitoes. Please indicate that even tough there were differences for your sample population, these were not statistically significant when tested for mosquitoes in general.
20. On line 454, please change "slightly" to "slightly but not significantly".
21. On line 460, please change "initiaed" to "initiated".
22. On line 531, you state that "SD+ALAN mosquitoes had higher glycogen levels than SD mosquitoes". This comparison was not statistically significant, please indicate that.
23. On line 588, please delete one copy of "involved".
24. On line 608, you state that "In short day-reared mosquitoes, light pollution led to slight increases in locomotor activity and protein levels in female mosquitoes...". These comparisons were not significant, please acknowledge that.
25. On line 596, change "reduceds" to "reduced".
26. On line 599, change "sa" to "as".
27. On line 620, change "present" to "presence".
28. On line 623, please add "in the absence of ALAN" at the end of the first half of the sentence after (A) ending with "SD mosquitoes".
29. On line 626, add "significantly" between "ALAN" and "suppressed".
30. On line 627, add "but not in SD conditions" after "in LD conditions".
31. On line 628, add "in the absence of ALAN" at then end of the first half of the sentence after (E) ending with "SD mosquitoes".
32. On line 628, add "significantly" between "ALAN" and "suppressed".
33. On line 629, add "but not in SD conditions" after "in LD conditions". You can also add "There was no longer a significant difference between LD and SD mosquitoes in the presence of ALAN.
Author Response
Thank you very much for your additional feedback! Please see the attachment for a point-by-point response to each comment.
